# GWAS meta-analysis reveals key risk loci in essential tremor pathogenesis
Astros Th. Skuladottir [1,2] ✉, Lilja Stefansdottir[1], Gisli H. Halldorsson [1], Olafur A. Stefansson [1], Anna Bjornsdottir[3], Palmi Jonsson[2,4], Vala Palmadottir[5], Thorgeir E. Thorgeirsson[1], G. Bragi Walters [1], Rosa S. Gisladottir[1,6], Gyda Bjornsdottir [1], Gudrun A. Jonsdottir[1], Patrick Sulem [1], Daniel F. Gudbjartsson [1,7], Kirk U. Knowlton[8], David A. Jones[9], Aigar Ottas[10], Estonian Biobank*, Ole B. Pedersen [11,12], Maria Didriksen [13], Søren Brunak [14], Karina Banasik [14], Thomas Folkmann Hansen [15], Christian Erikstrup [16,17], DBDS Genomic Consortium*, Jan Haavik [18,19], Ole A. Andreassen[20,21], David Rye[22], Jannicke Igland[23,24], Sisse Rye Ostrowski [12,13], Lili A. Milani [10], Lincoln D. Nadauld[9,25], Hreinn Stefansson [1] & Kari Stefansson [1,2] ✉

Essential tremor (ET) is a prevalent neurological disorder with a largely unknown underlying biology. In this genome-wide association study meta-analysis, comprising 16,480 ET cases and 1,936,173 controls from seven datasets, we identify 12 sequence variants at 11 loci. Evaluating mRNA expression, splicing, plasma protein levels, and coding effects, we highlight seven putative causal genes at these loci, including *CA3* and *CPLX1*. *CA3* encodes Carbonic Anhydrase III and carbonic anhydrase inhibitors have been shown to decrease tremors. *CPLX1*, encoding Complexin-1, regulates neurotransmitter release. Through gene-set enrichment analysis, we identify a significant association with specific cell types, including dopaminergic and GABAergic neurons, as well as biological processes like Rho GTPase signaling. Genetic correlation analyses reveals a positive association between ET and Parkinson's disease, depression, and anxiety-related phenotypes. This research uncovers risk loci, enhancing our knowledge of the complex genetics of this common but poorly understood disorder, and highlights *CA3* and *CPLX1* as potential therapeutic targets.

Essential tremor (ET) is one of the most common neurological disorders, affecting up to 5% of the population[1]. However, epidemiological studies show that the prevalence of the disorder is considerably underestimated as mildly affected individuals may not seek medical care[2].

ET is an isolated syndrome of bilateral upper limb postural or kinetic tremor, that may be with or without tremor of head, voice, or lower limbs and without other neurological signs such as dystonia, ataxia, or parkinsonism[3]. Although not life-threatening, the disorder can severely impact daily activities, reducing quality of life. Increasing age, European descent[4–6] and family history[7,8] are considered risk factors for ET.

Diagnosing ET can be challenging and often requires subspecialty consultation with a movement disorders neurologist. The diagnosis involves reviewing medical and family history and conducting a thorough neurological examination, as a biomarker or diagnostic test is not available[9]. Although there is no cure for ET, several treatment options are available to ease the symptoms. These include drug therapy (beta blockers, anti-epileptics, and tranquilizers), deep brain stimulation, and lifestyle modifications, such as avoiding triggers that can increase the severity of the tremors.

The cause of ET is not fully understood, but there is a growing support for the etiology of ET being partly related to abnormalities of the cerebello-thalamo-cortical network, including loss of Purkinje cells and reduced γ-aminobutyric acid (GABA) receptor expression in the dentate nucleus[10,11].

Previously, the largest genome-wide association study (GWAS) meta-analysis of ET reported five risk loci using data from 7177 cases and 475,877 controls from European populations[12]. Here, we more than double the ET case number, combining 16,480 cases and 1,936,173 controls in a GWAS meta-analysis and find 12 independent sequence variants at 11 loci, of which 8 are novel. Our research presents new genetic revelation regarding GABAergic dysfunction in ET, highlights the role of dopaminergic neurons, and provides further insight into the genetics of ET, offering clues that may lead to novel future treatment options.

## Results

### GWAS meta-analysis

In a meta-analysis of ET, we combined GWAS results from Iceland, Denmark, Estonia, Norway, UK, and USA (seven datasets) with summary statistics from a reported GWAS[12], resulting in 16,480 cases and 1,936,173 controls (Fig. 1 and Supplementary Data 1). Using a fixed-effect inverse variance model, we tested for association, under an additive model, between ET and sequence variants with imputation information over 0.8 and minor allele frequency (MAF) over 0.01% in each dataset (except the Estonian dataset and the reported GWAS, where variants with MAF over 1% were included). To account for multiple testing, we used weighted genome-wide significance thresholds based on the predicted functional impacts of the associated variants (Supplementary Data 2).

We uncovered association with ET at 11 loci (Supplementary Fig. 1 and Supplementary Data 3). Conditional analysis revealed a secondary signal at one of the loci (Supplementary Data 4). In total, we uncovered 12 independent common variants, 8 of which are novel (Fig. 2). There was no evidence of heterogeneity (all $P$-het > 0.05), indicating consistency of effects across the datasets (Supplementary Data 3). Five variants have been reported to associate with ET[12] and we show supportive evidence for all except one, at chromosome 1p13.1 (Supplementary Data 5). We report these signals directly or through a correlated variant ($r^2 \geq 0.8$) at the same locus (Supplementary Data 5). At chromosome 4p15.2, we observed a previously reported variant as the primary signal, along with a novel variant at the same locus acting as the secondary signal ($r^2 = 0.022$, Supplementary Fig. 1c, f).

### Potential causal genes

We searched for causal genes at the ET loci by evaluating the affected amino acid sequence of the lead variants and highly correlated variants ($r^2 \geq 0.8$), mRNA expression (expression quantitative trait loci [eQTLs]), splicing quantitative loci (sQTLs), and plasma protein levels (pQTLs) (Fig. 1). We found coding variants at three of the ET associated loci, in *CA3* (p.V31I, $r^2 = 1.00$ with lead variant), *EHBP1* (p.K720Q/K755Q, $r^2 = 0.96$ with lead variant), and *GCKR* (p.L446P, lead variant) (Fig. 3 and Supplementary Data 6). We found eQTLs for *BACE2*, *CPLX1*, *OTX1*, *C2orf16*, and *CA3* (Supplementary Data 7) and pQTLs for *CA3* (Supplementary Data 8) and *GCKR* (Supplementary Data 9).

The lead ET variant at the *CA3* locus confers protection against ET (rs955007-C, $P = 1.4 \times 10^{-12}$, OR = 0.92, Fig. 3. and Supplementary Data 3)

and is also highly correlated ($r^2 = 0.97$) with the primary *cis*-eQTL which decreases *CA3* expression in skeletal muscles (rs10088136-A, $P = 8.7 \times 10^{-13}$, $\beta = -0.13$, Supplementary Data 7). Using COLOC[13], we estimated that the posterior probability that the ET association and the eQTL are caused by the same variant is 89%. In addition, the lead variant is highly correlated ($r^2 = 1.00$) with the primary pQTL for carbonic anhydrase III and decreases its plasma levels (chr8:85445533, $P = 3.2 \times 10^{-131}$, $\beta = -0.22$, Supplementary Data 8). Furthermore, rs955007-C also associates with lower plasma levels of carbonic anhydrase XIII (located roughly 200KB upstream, $P = 6.3 \times 10^{-195}$, $\beta = -0.23$). However, rs955007 is not in high LD ($r^2 = 0.14$) with the primary pQTL at the region (Supplementary Data 8).

The lead intronic ET variant within *EHBP1* is highly correlated ($r^2 = 0.91$) with the top *cis*-eQTLs for *OTX1* in whole blood (rs76298426-C, $P = 10^{-1673}$, $\beta = 1.15$) and neutrophiles (rs146236066-CT, $P = 1.6 \times 10^{-51}$, $\beta = 0.98$, Supplementary Data 7).

We identified an intronic variant in *CPLX1* that confers risk of ET (rs13128363-T, $P = 9.0 \times 10^{-15}$, OR = 1.14, Fig. 3 and Supplementary Data 3) and is the top *cis*-eQTL for the gene in whole blood (Supplementary Data 7).

There were no sQTLs at the loci in whole blood.

Gene-set enrichment analysis in FUMA[14] highlighted cell types in the human embryonic midbrain such as dopaminergic neurons, GABAergic neuroblasts and neurons, and mediolateral neuroblast (Supplementary Data 10). In addition, the analysis revealed enrichment for pathways such as regulation of response to stress, cell adhesion, and Rho GTPase cycles (Supplementary Data 10).

The sex ratio in our study is close to 50% (52.9% females, Supplementary Data 1). When applying sex-specific models to the Icelandic, Danish, Norwegian, UK, and US-INTMT datasets for the 12 ET variants, none of the variants had an effect that significantly differed between the sexes, after accounting for multiple testing ($P$-het > 0.05/12 = 0.0042, Supplementary Data 11).

### Familial clustering and genetic variance explained

A close to complete genealogy exists for the Icelandic dataset. We did not find high-impact variants, including start-lost, stop-gain, stop-lost, splice donor, splice acceptor, or frameshift, segregating among 53 large families with high incidence of ET ($N \geq 5$). In addition, we did not see a significant difference in effects of 11 of the 12 lead variants between familial ($N = 1153$) and sporadic cases ($P$-het > 0.05/12 = 0.0042, Supplementary Fig. 3).

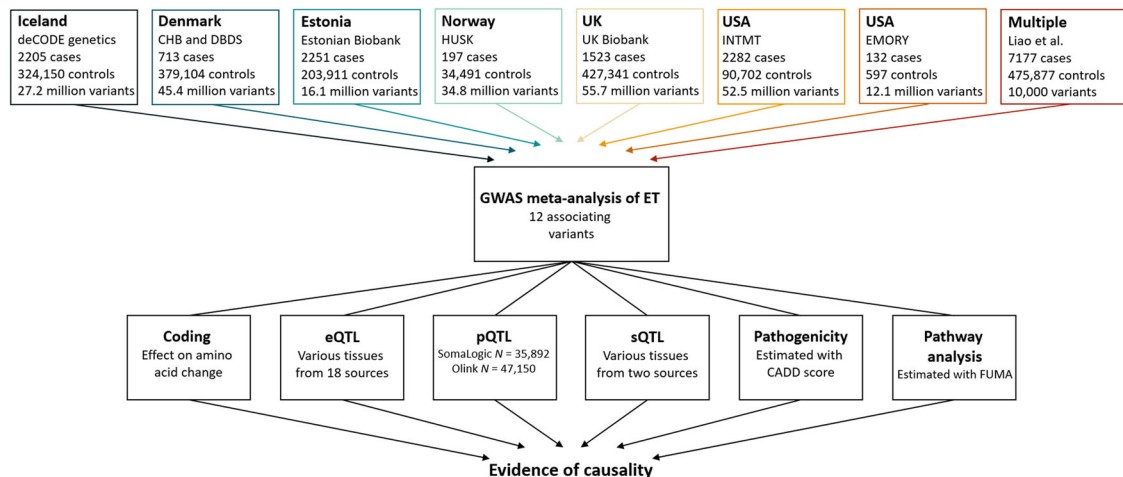

**Fig. 1 | Study design.** The first row lists the datasets used in the GWAS meta-analysis, number of ET cases, controls and variants analyzed. We included variants with MAF > 0.01% in all datasets except for the Estonian dataset and the previous GWAS[12], where variants with MAF > 1% were included. The summary data from a previous GWAS, only includes the top 10,000 variants. The last row lists the multiomics approaches used to search for potential causal genes. Expression quantitative

trait loci (eQTL) data sources are listed in Supplementary Data 15. Plasma protein levels (pQTL) were measured in Icelandic samples using Somalogic platform and in UK samples using the Olink platform. Splicing quantitative loci (sQTL) data were estimated using Icelandic RNA sequencing data, in addition to data imported from GTEx.

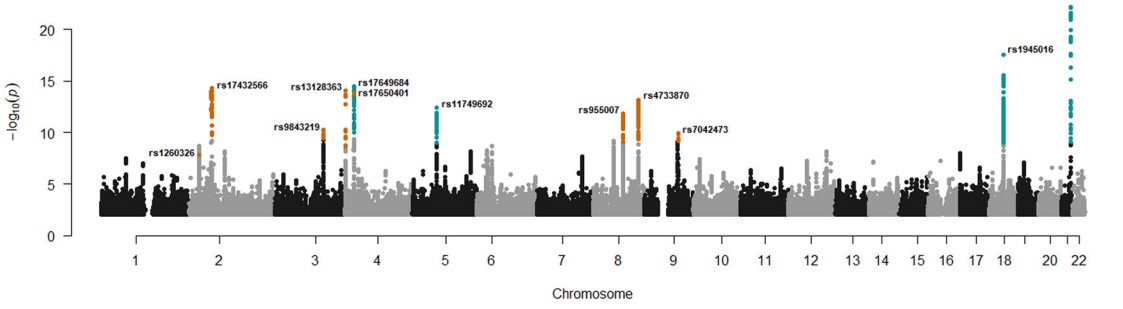

**Fig. 2 | Manhattan plot showing common variants in the ET meta-analysis.** The -log₁₀*P*-values (y-axis) are plotted for each variant against their chromosomal position (x-axis). Variants with *P*-values below their weighted variant-class threshold are highlighted. Novel variants are marked in orange and previously reported variants are marked in blue. *P*-values are two-sided and derived from a likelihood-ratio test. Manhattan plots for each dataset are shown in Supplementary Fig. 2.

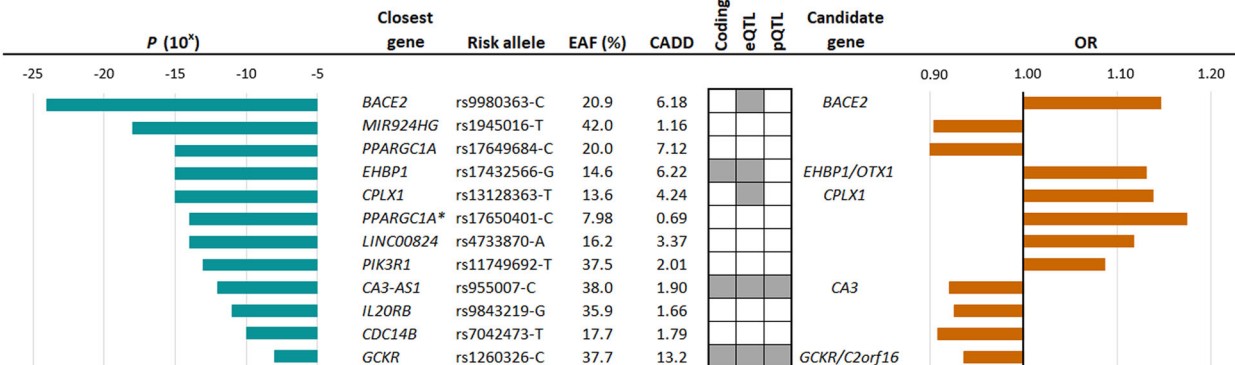

**Fig. 3 | Sequence variants that associate with ET and multiomics approaches used to uncover candidate causal genes.** Using multiomics approaches of the lead 12 variants, we identified 7 potential causal genes. Gray boxes indicate where data points to a candidate causal gene. Effects are shown for the minor allele. Combined Annotation Dependent Depletion (CADD)[84] score estimates the deleteriousness of sequence variants. Variants are considered pathogenic if CADD > 12.37. *Secondary signal at *PPARGC1A*.

Using the 12 independent variants, we estimated the genetic variance explained to be 4.4% (Supplementary Data 12).

## Genetic correlation

Considering the epidemiology and positive genetic correlation that has been reported between ET and Parkinson's disease (PD) and depression[12], we estimated the genetic correlation between ET and these two phenotypes using the most recent GWASs and cross-trait LD score regression. In line with previous reports, we observed a positive genetic correlation between ET and these phenotypes (PD[15], $r_g = 0.28$, $P = 1.1 \times 10^{-6}$; depression[16], $r_g = 0.15$, $P = 3.4 \times 10^{-5}$, Supplementary Data 13). In addition, we estimated the genetic correlation between ET and summary data from 1142 published GWASs (*P*-threshold $\leq 0.05/1142 = 4.4 \times 10^{-5}$) and found that ET correlates most strongly with anxiety-related phenotypes (e.g., feeling nervous, $r_g = 0.20$, $P = 2.0 \times 10^{-6}$, Supplementary Data 13).

## Discussion

We report a GWAS meta-analysis of ET that combines 16,480 cases and expands results from previous GWASs by identifying 12 variants at 11 loci, of which 8 are novel. We leveraged mRNA expression, including splicing, plasma protein measurements and predicted coding effects to highlight seven putative causal genes and the biological roles of some of the variants. Through gene-set enrichment analysis, we underscored the involvement of dopaminergic and GABAergic neurons in ET, as well as the biological significance of the Rho GTPase cycle. We did not find high-impact variants segregating in families with high prevalence of ET. We showed a positive genetic correlation between ET and PD, depression, and anxiety-related phenotypes.

Based on functional annotation, we highlighted seven genes that may participate in the pathogenesis of ET. One of the candidate causal genes is *CA3* which encodes carbonic anhydrase III and is in close proximity (>1 Mb) to other carbonic anhydrase genes including *CA1*, *CA2*, and *CA13*. Carbonic anhydrase inhibitors represent a class of drugs that have demonstrated the ability to improve tremors, potentially through modulating brain pH levels. Additionally, acidification facilitates GABA receptor potentiation which may facilitate the effect of the inhibitors, on tremor and seizures[17,18]. Primidone, an anti-epileptic drug widely used to treat ET patients, has been shown to inhibit carbonic anhydrase II[18]. The lead variant at the *CA3* locus confers protection against ET and is highly correlated with a variant that associates (top *cis*-eQTLs) with decreased expression of *CA3* in skeletal muscles. The variant also associates with decreased plasma levels of carbonic anhydrase III and XIII. The high LD at the locus suggest a potentially shared biological mechanism or pathway through which the variants exert their effects. Further research is needed to identify the most likely causal variants. Carbonic anhydrases are relevant proteins that may have an important role in the biology of ET. This notable finding lends support to the hypothesis that inhibiting carbonic anhydrases could lead to improved tremor control. Thus, targeting the interplay of *CA3* and its closely related enzymes might lead to the development of more targeted and effective treatments for individuals suffering from ET.

While ET may not be directly caused by imbalances in neurotransmitters, certain neurotransmitters have been implicated in its

development and severity, such as GABA[10,19,20]. *CPLX1* encodes Complexin-1, a soluble presynaptic protein that specifically enhances transmitter release by increasing fusogenicity of synaptic vesicles[21]. *CPLX1* is overexpressed in substantia nigra from PD patients[22] and dysregulation of the gene have been associated with neurogenetic disorders[23], including myoclonic epilepsy[24]. In addition, homozygous *Cplx1* knockout mice have the earliest known onset of ataxia seen in a mouse model[21,25,26]. We identified an intronic variant in *CPLX1* that increases the risk of ET and is the top *cis*-eQTL for *CPLX1* in blood. This finding aligns with previous research and strongly suggests the involvement of *CPLX1* in the pathogenicity of ET.

*OTX1* is a homeodomain transcription factor and is encoded by Orthodenticle homeobox 1. In the mammalian brain, *OTX1* is expressed in the forebrain and midbrain during early stages of neural development[27] and at later stages and adulthood at high levels in layers 5 and 6, the deepest layers of the cortical plate[28,29]. Layer 5 neurons convey signals controlling motor behavior via their projections to the colliculi, pons, and spinal cord[30–33]. *Otx1* mutant mice have been reported to show spontaneous epileptic behavior and multiple abnormalities affecting certain brain regions[34]. *Otx1* mutant animal studies suggest that Otx1 is required for the development of normal axonal connectivity and the generation of coordinated motor behavior[29]. The ET associated variant in *EHBP1* is in high LD with a missense variant in the same gene and is, additionally, highly correlated with two intronic variants in *EHBP1* and one variant in *OTX1*, all of which are top *cis*-eQTLs for *OTX1* expression in blood and neutrophiles. Based on these findings, our observations suggest that the intronic variant in *EHBP1* may contribute to the risk of ET by potentially upregulating the expression of *OTX1*. Consequently, *OTX1* emerges as a promising candidate gene that could play a significant role in the underlying pathogenic mechanism of ET. However, it is important to note that we did not investigate protein levels of OTX1, as it is neither measured on the Somalogic nor Olink platforms.

*GCKR* has been associated with high serum uric acid and purine metabolism disorders[35]. Extrapyramidal signs such as tremor are often observed in these disorders[36]. Individuals with high serum uric acid might experience tremors attributed to these metabolic issues, yet, due to the lack of routine serum uric acid testing, they could be incorrectly diagnosed with ET. Whether the tremor associated with the *GCKR* variant is a phenocopy or typical ET needs to be investigated further.

GABAergic dysfunction, consistently observed in ET patients[10,19,37], is a focal point of research, but its genetic underpinnings remain unidentified. Our gene-set enrichment analysis reinforces the proposed role of GABA in ET, yet the exact role of GABA in the causative framework requires further elucidation. In addition, our analysis found enrichment for dopaminergic neurons, a crucial regulator of extrapyramidal movement. Given the shared phenotypic and genetic traits between ET and PD, the association with dopaminergic neurons is anticipated, particularly as their selective degeneration in the substantia nigra pars compacta characterizes PD[38]. Notably, neurologist have long suspected a potential link between ET and PD, but definitive evidence for this relationship has remained elusive[39]. Furthermore, our analysis underscores the significance of various biological processes, most notably the Rho GTPase cycle. Rho GTPases regulate the actin cytoskeleton of dopaminergic neurons, thus influencing their degeneration[40,41], and have been implicated in PD[41,42]. Interestingly, our findings underscore the significance of stress response regulation, especially given the genetic overlap between ET and anxiety phenotypes. ET patients exhibit intensified tremors under stress, and anti-anxiety medications, such as Clonazepam, prove highly effective in alleviating these tremors.

Familial clustering of ET is well recognized. However, estimates of the proportion with a family history is highly debated and ranges from as low as 17% to as high as 100%[43]. Linkage studies have identified susceptibility loci on 3q13[44] and 2p24.1[45] but others have shown absence of linkage at these loci[46]. The absence of a significant difference between the effects of sporadic and familial cases and the lack of high-impact variants segregating in several large Icelandic families, underscores the potential that ET may be predominantly influenced by common variants or a combination of such variants, rather than rare variants. To gain deeper insights into the

contribution of common variants on ET risk, the creation of a polygenic risk score would be valuable. A PRS analysis, if conducted with larger and more diverse datasets, could offer further clarity on the polygenic nature of ET and thus, the assembly of larger cohorts in future research holds promise to increasing our understanding of the complex genetics of ET. Another limitation lies in the lack of ethnic diversity within the datasets studied. As an increasing amount of genotypic and phenotypic data becomes available for diverse ethnic backgrounds, the inclusion of greater diversity in future studies could be helpful in uncovering ethnicity-specific genetic contributions and advancing our understanding of the genetic underpinnings of ET.

In the previous GWAS conducted on ET, a subset consisting of 216 cases from the UK Biobank ET data was used. Regrettably, we could not ascertain whether these particular cases overlap with the UK Biobank cases utilized in our current study. It is worth mentioning that even if all of these cases overlap with our data, they represent only 2% of the total 9303 cases (excluding cases from the previous GWAS). Given their small proportion, the exclusion of these cases would not significantly impact the results or conclusions of our study.

To conclude, through a comprehensive GWAS meta-analysis and a multiomics approach using a substantial cohort, we have advanced our understanding of the genetics and pathogenesis of ET. This progress not only enhances our knowledge of this complex and prevalent neurological disorder but can also form the basis for future investigations into treatment strategies and personalized interventions.

## Methods
### Study sample and ethics statement
In this study, ET cases were defined using International Classification of Diseases 10 (ICD-10) code G25.0 or ICD-9 code 333.1 in all datasets, as described in detail below, in addition to the sample defined by Liao et al.[12]. All ethical regulations relevant to human research participants were followed. The data used in the GWAS meta-analysis were collected through studies approved by ethics committees governing each dataset and written informed consent was obtained from all participants. Personal identifiers of participants' data were encrypted for privacy protection purposes in accordance with the regulation in each country. Genetic ancestry quality control was performed for all datasets[47–50] and participants were genotypically verified as being of European descent. In total, we studied data from 16,480 ET cases and 1,936,173 controls (Fig. 1 and Supplementary Data 1).

**Iceland – deCODE genetics.** A large fraction of the Icelandic population has participated in a research program at deCODE genetics. Participants donated blood or buccal samples after signing a broad informed consent allowing the use of their samples and data in various projects approved by the National Bioethics Committee (NBC). The data in this study was approved by the NBC (VSN-17-142-V5; VSNb2017060004/03.01) following review by the Icelandic Data Protection Authority. All personal identifiers of the participants' data were encrypted in accordance with the regulations of the Icelandic Data Protection Authority. The Icelandic ET cases were identified from medical records, filed from 1985 to 2022, through collaboration with physicians at Landspitali—National University Hospital in Reykjavik, the Registry of Primary Health Care Contacts, and the Registry of Contacts with Medical Specialists in Private Practice.

**Denmark – The Copenhagen Hospital Biobank and The Danish Blood Donor Study.** The Copenhagen Hospital Biobank (CHB) is a research biobank, which contains samples obtained during diagnostic procedures on hospitalized and outpatients in the Danish Capital Region hospitals. Data analysis was performed under the Developing the basis for personalized medicine in degenerative and episodic brain disorders protocol, approved by the National Committee on Health Research Ethics (H-21058057). The Danish Blood Donor Study (DBDS) Genomic Cohort is a nationwide study of ~160,000 blood donors[51]. The Danish Data Protection Agency (P-2019-99) and the National Committee on

Health Research Ethics (NVK-1700407) approved the studies under which data on DBDS participants were obtained. The DBDS data requested for this study was approved by the DBDS steering committee.

**Estonia – Estonian Biobank.** The Estonian Biobank is a population-based cohort of approximately 210,000 participants, each accompanied by a variety of phenotypic and health-related data[52]. Upon recruitment, participants granted permission through signed consents for subsequent linkage to their electronic health records, enabling the longitudinal accumulation of phenotypic details. The Estonian Biobank facilitates access to the records from the National Health Insurance Fund Treatment Bills (since 2004), Tartu University Hospital (since 2008), and North Estonia Medical Center (since 2005). For each participant, data is available on diagnoses coded in ICD-10 and drug dispensing records, including ATC codes, prescription statuses, and purchase dates (when available). The activities of the EstBB are regulated by the Human Genes Research Act, which was adopted in 2000 specifically for the operations of the EstBB. Analysis of individual level data from the EstBB was carried out under ethical approval 1.1-12/624 from the Estonian Committee on Bioethics and Human Research (Estonian Ministry of Social Affairs), using data according to release application [6-7/GI/29 977] from the Estonian Biobank.

**Norway – The Hordaland Health Study.** The Hordaland Health Study (HUSK) is a community-based study in Western Norway conducted as a collaboration between the University of Bergen, the Norwegian Health Screening Service and the Municipal Health Service in Hordaland (https://husk-en.w.uib.no/)[53]. In 1992–93 and 1997–99 participants were invited based on year of birth and site of residence. Residents from Hordaland County born 1950–52 and residents from Bergen and three neighboring municipalities born 1925–27, in addition to a random sample born 1926–49 were invited in 1992–93. In 1997–99, previous participants born 1950–51 and 1925–27 were reinvited, in addition to all residents in Hordaland County born 1953–57. In total, approximately 36,000 individuals participated in the study, 18,000 in 1992–93 and 26,000 in 1997–99, with some participating at both times. ET cases were identified through diagnostic codes reported in the patient registry during 2008 to 2021. The HUSKment study is approved by the Regional Committee for Medical Research Ethics Western Norway, reference 2018/915.

**The UK – The UK Biobank.** The UK Biobank resource has collected extensive phenotype and genotype data from ~500,000 participants in the age range 40–69, from across the UK after signing an informed consent for the use of their data in genetic studies[54]. The North West Research Ethics Committee reviewed and approved UK Biobank's scientific protocol and operational procedures (REC Reference Number: 06/MRE08/65). This study was conducted using the UK Biobank resource under application number 42256. ET cases were identified in General Practice clinical event records (Field ID 42040) and UK hospital diagnoses (Field ID 41270 and 41271).

**The US – Intermountain Healthcare.** Participants, voluntary US residents over the age of 18 years, were recruited by The Intermountain Inspire Registry and The HerediGene: Population study[55], a large-scale collaboration between Intermountain Healthcare, deCODE genetics, and Amgen, Inc (https://intermountainhealthcare.org). The Intermountain Healthcare Institutional Review Board approved this study, and all participants provided written informed consent prior to enrollment.

**The US – Emory General Clinical Research Center.** The Clinical Research in Neurology (CRIN) provides an umbrella structure for subject enrollment in observational and genetic studies in neurology, consent-approved data sharing across studies and disorders, and consistent sample processing. Participants were recruited under the CRIN protocol through support from Emory Clinical Research Center NIH/NCRR M01 RR00039 (CRIN Infrastructure support). The study was approved by the Emory Institutional Review Board (IRB) and informed consent was obtained from all subjects. ET genotyping work was done under specific IRB protocols. Samples were drawn from either review of previously enrolled subjects in the CRIN database, or prospective enrollment of ET subjects into CRIN/ET observational and genetics work. All subjects underwent a basic structured interview for demographics and family history. A Folstein Mini Mental Status Exam was administered to all CRIN subjects by trained CRIN personnel supervised by a neuropsychologist per published guidelines. All CRIN database subjects enrolled prior to January 2007 with a reported diagnosis of 333.1 were reviewed. ET subjects were called in for full in-person assessments whenever possible. ET subjects mid-2006 onward were recruited through IRB-approved ads in the Emory Movement Disorders and Neurosurgery deep brain stimulation group clinics, and ET community education events. ET subjects and family members were examined directly by at least one movement disorders specialist; two independent exams were obtained whenever possible (a tremor rating scale derived from the Fahn-Tolosa-Marin scale and Tremor Research Group scale items, the motor United Parkinson Disease Rating Scale, Tinetti gait and balance scales[56], tandem gait[57], and assessment for dystonia). Semi-structured interviews included ET specific questions derived from the Fahn-Tolosa-Marin scale and WHIGET studies[58]. CRIN review and new enrollment subjects were given a research diagnosis of ET using Movement Disorders Society and Tremor Research Group criteria. ET cases with either Parkinson's disease or dystonia were excluded. Subjects were excluded based on a number of criteria; if an in-person exam and re-interview determined a different diagnosis, if movement disorders clinical notes listed an uncertain or different final diagnosis (i.e., medication induced tremor), if there was an incomplete examination, lack of medication response, or other data to clearly establish an ET research diagnosis.

## Genotyping and imputation

**Iceland – deCODE genetics.** The genomes of 63,460 Icelanders were whole genome sequenced (WGS)[47,59] using GAIIx, HiSeq, HiSeqX, and NovaSeq Illumina technology to a mean depth of 38×. Genotypes of single nucleotide polymorphisms (SNPs) and insertions/deletions (indels) were identified and called jointly with Graphtyper[60,61]. Over 173,000 Icelanders (including all WGS Icelanders) were genotyped using various Illumina SNP arrays[47,59]. The genotypes were long-range phased[62], which allows for improving genotype calls using haplotype sharing information. Subsequently, extensive encrypted genealogic information was used to impute variants into the chip-typed Icelanders, as well as ungenotyped close relatives[63] to increase the sample size and power for association analysis.

**Denmark – The Copenhagen Hospital Biobank and The Danish Blood Donor Study.** The Danish samples from the CHB and DBDS were genotyped using Illumina Global Screening Array, and long-range phased together with 270,627 genotyped samples from North-western Europe using Eagle2[64]. Samples and variants with less than 98% yield were excluded. A haplotype reference panel was prepared in the same manner as for the Icelandic data[47,62] by phasing genotypes of 25,215 WGS individuals (sequenced with NovaSeq Illumina technology to a mean depth of 20×) from North-western Europe, including 8,360 Danes, using the phased chip data. Graphtyper[60,61] was used to call the genotypes which were subsequently imputed into the phased chip data. WGS, chip-typing, quality control, long-range phasing, and imputation from which the data for this analysis were generated was performed at deCODE genetics.

**Estonia – Estonian Biobank.** The samples from the Estonian Biobank were genotyped at the Genotyping Core Facility of the Institute of Genomics at the University of Tartu, using the Illumina Global Screening

Array. In total, 212,955 samples passed quality control. Samples were excluded from the analysis if their call-rate was below 95% or if the gender, identified by the heterozygosity of the X chromosome, did not align with the gender documented in phenotype data. Variants were excluded if the call-rate was below 95% or if the HWE p-value was less than 1e-4 (only autosomal variants). In addition, SNPs that showed potential traces of batch bias were removed. Two batch bias control steps were performed: 1) SNPs that showed poor cluster separation results among any of Estonian Biobank genotyping experiments were removed. The threshold for SNP removal was Illumina GenTrain score <0.6 and/or cluster separation score <0.4. 2) SNPs that showed inconsistent allele frequency among genotyping experiments were excluded. First, allele frequency was calculated for each SNP for each genotyping experiment with more than 10,000 samples. Next, mean allele frequency was calculated. Finally, if SNP allele frequency was more than 5% away from the mean in any of genotyping experiments, the SNP was excluded from the merged dataset. Prior to imputation, variants with MAF less than 1% and indels were removed. The Eagle v2.4.1[64] was used for prephasing and imputation was executed using Beagle v5.4 (beagle.22Jul22.46e)[65]. An imputation reference, specific to the Estonian population, consisting of 2056 WGS samples was used[66]. Participants with non-European assigned group ancestry were removed, leaving a total of 206,162 samples.

**Norway – The Hordaland Health Study.** The Norwegian dataset was genotyped using Illumina SNP arrays (either OmniExpress or Global Screening Array). The chip-genotyping quality control and imputation were performed at deCODE genetics, where the same methods used for the Danish sample were applied. The imputation process relied on the same haplotype reference panel as the Danish sample, a panel composed of phased genotypes of 25,215 WGS samples of European ancestry, including 3,336 samples of Norwegian origin.

**The UK – The UK Biobank.** The samples from the UK Biobank were genotyped using two different Affymetrix chips – the UK BiLEVE Axiom in the first 50,000 individuals[67], and the Affymetrix UK Biobank Axiom array[68] in the remaining participants. In total, 428,864 participants have been genotyped and 131,272 WGS. Samples with variant yield below 98% were filtered out and any duplicate samples were removed. High-quality sequence variants and indels to a mean depth of at least 20× were identified using Graphtyper[60,61]. Quality-controlled chip-genotype data were phased using Shapeit4[69] and variants where at least 50% of the samples had a genotype quality score above 0 were used to prepare a haplotype reference panel using in-house tools and the long-range phased chip data. The variants in the haplotype reference panel were imputed into the chip-genotyped samples using the same in-house tools and methods described for the Icelandic data[47,62].

**The US – Intermountain Healthcare.** The Intermountain dataset was genotyped using Illumina Global Screening Array chips ($N = 76,660$) and WGS with NovaSeq Illumina technology ($N = 20,632$). The samples were filtered on 98% variant yield and duplicates removed. High-quality sequence variants and indels with at least a mean depth of 20× were identified with Graphtyper[60,61]. Quality-controlled genotype data were phased with Shapeit4[69]. A phased haplotype reference panel was prepared with the same in-house tools and methods described for the Icelandic data[47,62].

**The US – Emory General Clinical Research Center.** The genotyping of the Emory dataset has been described previously[50]. In short, the Emory dataset was genotyped using three types of chips from Illumina (HumanHap300, HumanHap300-Duo and HumanCNV370-Duo). These chips have 314,125 SNPs in common. Prior to analysis, certain SNPs were excluded based on the following criteria; being monomorphic, having less than 95% yield in either cases or controls, deviating from Hardy-Weinberg equilibrium, or displaying divergent allele frequencies

between the chips. Additionally, samples with a call-rate less than 98% were excluded.

## Statistics and reproducibility

We applied logistic regression assuming an additive model using the expected allele counts as covariates, and combined the results with the available GWAS summary statistics of 10,000 variants[12] to test for association between sequence variants and ET. The covariates we used in the association analysis are described in Supplementary Data 14 for the datasets. We used LD score regression to account for distribution inflation due to cryptic relatedness and population stratification[70] and used the intercepts as correction factors.

We combined the results from the association analysis of all of the datasets together with the summary statistics (only the top 10,000 variants)[12] using a fixed-effects inverse variance method[71] based on effect estimates and standard errors in which each dataset was assumed to have a common OR but allowed to have different population frequencies for alleles and genotypes. Sequence variants were mapped to NCBI Build38 and matched on position and alleles to harmonize the datasets. The genome-wide significance threshold was corrected for multiple testing using a weighted Bonferroni adjustment that controls for the family-wise error rate. Variants were weighted based on predicted functional impact[72] (Supplementary Data 2).

In a random-effects method, a likelihood ratio test was performed in all genome-wide associations to test the heterogeneity of the effect estimate in the datasets; the null hypothesis is that the effects are the same in all datasets and the alternative hypothesis is that the effects differ between datasets.

The primary signal at each locus was defined as the sequence variant with the lowest Bonferroni-adjusted *P*-value using the adjusted significance thresholds (Supplementary Data 2). To identify secondary signals at each locus (defined as 1 Mb from the index variants), we performed conditional association analyses using the true imputed genotype data of each dataset except the Estonian and US-EMORY datasets and the summary statistics where an approximate conditional analysis implemented in the GCTA software[73] was used. LD between variants was estimated using a set of 5,000 WGS Icelanders. After adjusting for all variants in high LD ($r^2 > 0.8$) and vice versa, the *P*-values were combined for all datasets to identify the most likely causal variant at each locus and any secondary signals. Based on the number of variants tested, we chose a conservative *P*-value threshold of $<5 \times 10^{-8}$ for secondary signals.

Manhattan plots were generated using the qqman package in R[74].

## Functional data

To highlight potential causal genes associating with ET, we annotated the variants associating with ET or variants in high LD ($r^2 \geq 0.8$ and within ± 1 Mb) that are predicted to affect coding or splicing of a protein (variant effect predictor using Refseq gene set), mRNA expression (top local expression quantitative trait loci [*cis*-eQTL]) in multiple tissues from deCODE, GTEx (https://gtexportal.org), and other public datasets (Supplementary Data 15), and/or plasma protein levels (top protein quantitative trait loci [pQTL]) in large proteomic datasets from Iceland and the UK.

RNA sequencing was performed on whole blood ($N = 17,848$) and subcutaneous adipose tissue ($N = 769$). RNA isolation was performed using RNAzol RT according to manufacturer's protocol (Molecular Research Center RN 190). We isolated RNA using Chemagic Total RNA Kit special (PerkinElmer) in whole blood and RNAzol RT in adipose tissue, according to the manufacturer's instructions (Molecular Research Center, RN190). The concentration and quality of the RNA were determined with an Agilent 2100 Bioanalyzer (Agilent Technologies). RNA was prepared and sequenced on the Illumina HiSeq 2500 and Illumina Novaseq systems according to the manufacturer's recommendation. RNA-seq reads were aligned to personalized genomes using the STAR software package v.2.5.3 with Ensembl v.87 gene annotations[75,76]. Gene expression was computed based on personalized transcript abundances using kallisto[77]. Association between sequence variants and gene expression (*cis*-eQTL) was estimated using a generalized linear regression, assuming additive genetic model and

quantile-normalized gene expression estimates, adjusting for measurements of sequencing artefacts, demographic variables, blood composition, and PCs[78]. The gene expression PCs were computed per chromosome using a leave-one-chromosome-out method.

Quantification of alternative RNA splicing in whole blood was done using LeafCutter[79]. The *cis* association between sequence variants and quantified splicing (*cis*-sQTL) was estimated using linear regression assuming an additive genetic model and quantile-normalized percentage-spliced-in (PSI) values of each splice junction, adjusting for measurements of sequencing artefacts, demography variables, and 15 leave-one-chromosome-out PCs of the quantile-normalized PSI matrix. All variants with MAF > 0.2% within 30 Kb of each LeafCutter cluster were tested.

Icelandic plasma samples were collected through two main projects: the Icelandic Cancer Project (52% of participants; samples collected from 2001 to 2005) and various genetic programs at deCODE genetics, mainly the population-based deCODE Health study. The average participant age was 55 years (SD = 17 years) and 57% were women. In the case of repeated samples for an individual, one was randomly selected. This left measurements for 39,155 individuals. Of these, 35,892 Icelanders were used in the protein GWASs, because they also had genotype information[80]. The plasma samples were measured with SomaScan v4 assay (SomaLogic®). The assay scanned 4,907 aptamers that measure 4719 proteins. Plasma protein levels were standardized and adjusted for year of birth, sex, and year of sample collection.

The plasma levels of a subset of 47,150 individuals in the UK Biobank were measured with the Olink Explore 1536 platform as a part of the UKB–Pharma Proteomics Project (UK Biobank application no. 65851)[81] at Olink's facilities in Uppsala, Sweden. The majority of the samples were randomly selected across the UK Biobank. Plasma protein levels were standardized to a normal distribution.

We performed gene-based enrichment analysis using the GENE2-FUNC tool in FUMA[14]. The genes on the loci (closest protein coding gene per locus was prioritized) that met traditional genome-wide significance ($P \leq 5 \times 10^{-8}$) in the ET meta-analysis were tested for over-representation in different gene sets, including Gene Ontology biological processes (MsigDB c5), Reactome (MsigDB c2) and Cell type signatures (MsigDB c8). A Bonferroni test was used for multiple comparison correction.

### Familial clustering

We used family-based method to test rare coding variants for segregation within a pedigree. We focused our search on rare (carried by <30 whole-genome-sequenced individuals), high-penetrance coding variants that could account for the familial clustering. To test for association, we created a scoring function based on the coding effect of the variant and its cosegregation with ET, inside and outside of the pedigrees, and used genome-wide simulations to estimate the significance. This method has been described in detail elsewhere[82].

### Estimation of genetic variance explained

We calculated the variance explained ($h^2$) using the β and EAF from the ET meta-analysis of each of the independent and significant variant with the formula $h^2 = \beta^2 \times (1\text{-EAF}) \times 2\text{EAF}$[83].

### Genetic correlation

Cross-trait LD score regression[70] was used to estimate the genetic correlation between the ET meta-analysis and GWAS meta-analyses of other neurological phenotypes, namely Parkinson's disease and major depressive disorder. We also estimated the genetic correlation between the ET meta-analysis and 1152 previously published GWAS traits ($P \leq 3.8 \times 10^{-5}$) each with an effective sample size over 5,000 for an unbiased estimate of genetic correlation and heritability. To avoid bias due to sample overlap, we excluded the UK dataset from the ET meta-analysis. We used results for about 1.2 million well imputed variants, and for LD information we used precomputed LD scores for European populations (downloaded from: https://data.broadinstitute.org/alkesgroup/LDSCORE/eur_w_ld_chr.tar.bz2).

### Reporting summary

Further information on research design is available in the Nature Portfolio Reporting Summary linked to this article.

## Data availability

The GWAS summary statistics for the ET meta-analysis are available at https://www.decode.com/summarydata/. Other data generated or analyzed in this study are included in the article and Supplementary data and information.

## Code availability

GraphTyper (v2.0-beta, GNU GPLv3 license) at https://github.com/DecodeGenetics/graphtyper

Svimmer (v0.1, GNU GPLv3 license), the structural variant merging software at https://github.com/DecodeGenetics/svimmer

SHAPEIT4 (v4.2.2) at https://odelaneau.github.io/shapeit4/

Eagle2 (v2.4.1) at http://www.hsph.harvard.edu/alkes-price/software/

Beagle (v5.4) at https://faculty.washington.edu/browning/beagle/beagle.html

GCTA (v1.93.3beta2) at https://yanglab.westlake.edu.cn/software/gcta/#Overview

STAR (v2.5.3) at http://star.mit.edu/

Kallisto at https://pachterlab.github.io/kallisto/

LeafCutter at https://davidaknowles.github.io/leafcutter/

LD score regression (first release) at https://github.com/bulik/ldsc

qqman package (v0.1.6) at https://github.com/stephenturner/qqman

Axiom genotyping algorithm (v1) at https://www.thermofisher.com/is/en/home.html

FUMA at https://fuma.ctglab.nl/

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

## Acknowledgements

We thank the participants in this study for their valuable contribution to research. We thank all investigators and colleagues who contributed to data collection, phenotypic characterization of clinical samples, genotyping, and analysis of the whole-genome association data. The Estonian dataset was funded by European Union through the European Regional Development Fund Project No. 2014-2020.4.01.15-0012 GENTRANSMED and by Estonian Research Council grant PRG1291. Data analysis of the Estonian dataset was carried out in part in the High-Performance Computing Center of University of Tartu. This research was conducted using the UK Biobank Resource (application number 42256).

## Author contributions

A.T.S., H.S., and K.S. designed the study. A.T.S., L.S., G.H.H., O.A.S., H.S., and K.S. analyzed the data and interpreted the results. Data collection and subject ascertainment and recruitment was carried out by A.T.S., A.B., P.J., V.P., and H.S. for the Icelandic dataset, L.A.M. and A.O. for the Estonian dataset, K.U.K., D.A.J., and L.D.N. for the US-INTMT dataset, J.I., J.H., and O.A.A. for the Norwegian dataset, D.R. for the US-EMORY dataset, and S.R.O., O.B.P., M.D., S.B., K.B., T.F.H., C.E. for the Danish dataset. A.T.S. drafted the manuscript with input and comments from G.H.H., A.B., G.B.W., R.S.G., G.B., T.E.T., G.A.J., P.S., D.F.G., S.R.O., L.A.M., J.H., O.A.A., J.I., H.S., and K.S. All authors read the final version of the manuscript.

## Competing interests

A.T.S., L.S., G.H.H., O.A.S., G.B.W., R.S.G., G.B., T.E.T., G.A.J., P.S., D.F.G., H.S., and K.S. are employees of deCODE genetics/Amgen Inc. The remaining authors declare no competing interests.

## Additional information

[1]deCODE genetics/Amgen Inc., Reykjavik, Iceland. [2]Faculty of Medicine, University of Iceland, Reykjavik, Iceland. [3]Heilsuklasinn Clinic, Reykjavik, Iceland. [4]Department of Geriatric Medicine, Landspitali University Hospital, Reykjavik, Iceland. [5]Department of Internal Medicine, Landspitali University Hospital, Reykjavik, Iceland. [6]Faculty of Icelandic and Comparative Cultural Studies, University of Iceland, Reykjavik, Iceland. [7]Faculty of Engineering and Natural Sciences, University of Iceland, Reykjavik, Iceland. [8]Intermountain Medical Center, Intermountain Heart Institute, Salt Lake City, USA. [9]Precision Genomics, Intermountain Healthcare, Saint George, Utah, UK. [10]Estonian Genome Centre, Institute of Genomics, University of Tartu, Tartu, Estonia. [11]Department of Clinical Immunology, Zealand University Hospital, Køge, Denmark. [12]Department of Clinical Medicine, Faculty of Health and Medical Sciences, University of Copenhagen, Copenhagen, Denmark. [13]Department of Clinical Immunology, Copenhagen University Hospital, Righospitale, Copenhagen, Denmark. [14]Novo Nordisk Foundation Center for Protein Research, Faculty of Health and Medical Sciences, University of Copenhagen, Copenhagen, Denmark. [15]Danish Headache Center, Department of Neurology, Copenhagen University Hospital, Righospitalet-Glostrup, Copenhagen, Denmark. [16]Department of Clinical Immunology, Aarhus University Hospital, Righospitalet, Copenhagen, Denmark. [17]Department of Clinical Medicine, Faculty of Health and Medical Sciences, Aarhus University, Aarhus, Denmark. [18]Department of Biomedicine, University of Bergen, Bergen, Norway. [19]Bergen Center of Brain Plasticity, Division of Psychiatry, Haukeland University Hospital, Bergen, Norway. [20]Institute of Clinical Medicine, University of Oslo, Oslo, Norway. [21]NORMENT, Division of Mental Health and Addiction, Oslo University Hospital, Oslo, Norway. [22]Emory Department of Neurology, Wesley Woods Health Center, Atlanta, GA, USA. [23]Department of Global Public Health and Primary Care, University of Bergen, Bergen, Norway. [24]Department of Health and Caring sciences, Western Norway University of Applied Sciences, Bergen, Norway. [25]Stanford University, School of Medicine, Stanford, CA, USA. *Lists of authors and their affiliations appear at the end of the paper. ✉e-mail: astros.skuladottir@decode.is; kstefans@decode.is

## Estonian Biobank

Tõnu Esko[10], Reedik Mägi[10], Mari Nelis[10] & Georgi Hudjashov[10]

## DBDS Genomic Consortium

Karina Banasik [14], Jakob Bay[11], Jens Kjærgaard Boldsen[16], Thorsten Brodersen[11], Søren Brunak [14], Kristoffer Burgdorf[14], Mona Ameri Chalmer[15], Maria Didriksen [13], Khoa Manh Dinh[16], Joseph Dowsett[13], Christian Erikstrup [16,17], Bjarke Feenstra[13,26], Frank Geller[13,26], Daniel Gudbjartsson[1,7], Thomas Folkmann Hansen [15], Lotte Hindhede[16], Henrik Hjalgrim[27], Rikke Louise Jacobsen[13], Gregor Jemec[28], Bitten Aagaard Jensen[29], Katrine Kaspersen[16], Bertram Dalskov Kjerulff[16], Lisette Kogelman[15], Margit Anita Hørup Larsen[13], Ioannis Louloudis[14], Agnete Lundgaard[14], Susan Mikkelsen[16], Christina Mikkelsen[13], Ioanna Nissen[13], Mette Nyegaard[30], Sisse Rye Ostrowski [12,13], Ole Birger Pedersen[11,12], Alexander Pil Henriksen[14], Palle Duun Rohde[30], Klaus Rostgaard[26,27], Michael Schwinn[13], Kari Stefansson [1,2]✉, Hreinn Stefánsson[1], Erik Sørensen[13], Unnur Þorsteinsdóttir[1,2], Lise Wegner Thørner[13], Mie Topholm Bruun[31], Henrik Ullum[32], Thomas Werge[12,33] & David Westergaard[14]

[26]Department of Epidemiology Research, Statens Serum Institut, Copenhagen, Denmark. [27]Danish Cancer Society Research Center, Copenhagen, Denmark. [28]Department of Dermatology, Zealand University hospital, Roskilde, Denmark. [29]Department of Clinical Immunology, Aalborg University Hospital, Aalborg, Denmark. [30]Department of Health Science and Technology, Faculty of Medicine, Aalborg University, Aalborg, Denmark. [31]Department of Clinical Immunology, Odense University Hospital, Odense, Denmark. [32]Statens Serum Institut, Copenhagen, Denmark. [33]Institute of Biological Psychiatry, Mental Health Centre, Sct. Hans, Copenhagen University Hospital, Roskilde, Denmark.

