## [Peer Review File · Communications Biology]

Reviewers' comments:

Reviewer #1 (Remarks to the Author):

The authors of "Multiomics analysis of essential tremor uncovers potential therapeutic targets" report some potentially interesting new genetic findings in a large meta-analysis of essential tremor. Overall I don't have many comments and think this is a good manuscript that pushes the ET field in the right direction, some suggestions below:

- For the CA3 locus qtl effect, worth to specify in the results section the direction of expression, does this goes up or down with risk ? Similar as the pQTL for this gene
- In addition why is chr8:85445533 reported like this and others have rs-numbers "rs955007 is not in high LD" worth to add what the LD really is here R2 and D'
- Finally what is the real conclusion here? Does the coding variant cause the effect or is the coding variant just by chance on the risk haplotype?
- The discussion is quite long overall.
- It looks like 4 GWAS regions are almost resolved, is there any idea what the other 7 might do and how they could be resolved?

Reviewer #2 (Remarks to the Author):

This study conducted a genome-wide association study (GWAS) on essential tremor (ET) in a large sample population from multiple centers. It identified new associated loci and candidate genes using multi-omics data, thereby expanding our understanding of the genetics of ET. The study's findings were very meaningful, but some aspects remained unclear in the article. Here are my suggestions:

1. The study is not a multi-omics study but a GWAS, so the title of the article is misleading.
2. The section on candidate gene identification mentions sQTLs but no corresponding results are presented.
3. The familial analysis section is too simplistic. Please expand on the methods and results sections, such as the number of analyzed pedigrees and the types of variants analyzed.
4. The genetic correlation section is similar to previously reported results. Please explain the unique aspects of this study.
5. The post-GWAS section highlights multiple signals indicating an association with ET for the GCKR gene. Please discuss the potential mechanisms of this gene in the discussion section.
6. The paper contains several errors, please carefully review and make corrections. For example, main text line 99, "frequency (MAF) over 0.01%"; figure 1, the sample size and the number of variants do not match with other parts of the manuscript; supplementary Figure 3, "for the 11 variants associated with ET".

Response to reviewer's comments

We appreciate insightful and constructive comments from the reviewers. We have modified the manuscript in line with their comments. Below are our detailed responses.

Reviewer #1

The authors of “Multiomics analysis of essential tremor uncovers potential therapeutic targets” report some potentially interesting new genetic findings in a large meta-analysis of essential tremor. Overall I don't have many comments and think this is a good manuscript that pushes the ET field in the right direction, some suggestions below.

- 1.1 For the CA3 locus qtl effect, worth to specify in the results section the direction of expression, does this goes up or down with risk ? Similar as the pQTL for this gene

Response (1.1)

In the manuscript, we mention the effect for the eQTL (rs10088136-A, $P = 8.7 \times 10^{-13}$, $\beta = -0.13$, Supplementary Data 7) and the pQTL (chr8:85445533, $P = 3.2 \times 10^{-131}$, $\beta = -0.22$, Supplementary Data 8). We have clarified by adding the words in bold (page 7):

“The lead ET variant at the *CA3* locus confers protection against ET (rs955007-C, $P = 1.4 \times 10^{-12}$, OR = 0.92, Fig 3. and Supplementary Data 3) and is also highly correlated ($r^2 = 0.97$) with the primary *cis*-eQTL **which decreases** *CA3* expression in skeletal muscles (rs10088136-A, $P = 8.7 \times 10^{-13}$, $\beta = -0.13$, Supplementary Data 7). ... In addition, the lead variant is highly correlated ($r^2 = 1.00$) with the primary pQTL for carbonic anhydrase III **and decreases its plasma levels** (chr8:85445533, $P = 3.2 \times 10^{-131}$, $\beta = -0.22$, Supplementary Data 8).”

- 1.2 In addition why is chr8:85445533 reported like this and others have rs-numbers

Response (1.2)

The top pQTL for carbonic anhydrase III is a multiallelic site that was called using GraphTyper¹. The reported allele is the reference allele and unfortunately it is highly unlikely that it will be submitted to dbSNP for an rsID.

- 1.3 “rs955007 is not in high LD” worth to add what the LD really is here R2 and D'

Response (1.3)

To maintain consistency with other reporting in the manuscript, we have included the r^2 value (page 7).

“However, rs955007 is not in high LD ($r^2 = 0.14$) with the primary pQTL at the region (Supplementary Data 8).”

- 1.4 Finally what is the real conclusion here? Does the coding variant cause the effect or is the coding variant just by chance on the risk haplotype?

Response (1.4)

We thank the reviewer for an insightful question. Our findings highlight a complex scenario at the locus, where the coding variant (p.Val31Ile) is strongly correlated with multiple other variants, all of which are associated with similar effects on ET. Importantly, among these correlated variants are the top eQTL and pQTL signals. The presence of the eQTL and pQTL provides a functional basis for understanding how these variants might impact ET, beyond the genetic association. Although p.Val31Ile is a candidate for causality, the high LD at the locus complicates the direct attribution of the ET effects to a single variant. This complexity underscores the necessity for further investigation. We have improved the manuscript by addressing the comment in Discussion (page 10).

“The high LD at the locus suggest a potentially shared biological mechanism or pathway through which the variants exert their effects. Further research is needed to identify the most likely causal variants.”

1.5 The discussion is quite long overall.

Response (1.5)

We agree with the reviewer. Nonetheless, our study touches on several novel aspects that have not been previously discussed. However, we have made efforts to condense the discussion section without removing important details.

1.6 It looks like 4 GWAS regions are almost resolved, is there any idea what the other 7 might do and how they could be resolved?

Response (1.6)

Larger ET GWAS studies are expected to reveal additional sequence variants conferring ET risk. A more comprehensive pathway analysis, incorporating a larger number of loci, will hopefully increase our understanding of how genes at the different GWAS regions may contribute to the development of ET.

Reviewer #2

This study conducted a genome-wide association study (GWAS) on essential tremor (ET) in a large sample population from multiple centers. It identified new associated loci and candidate genes using multi-omics data, thereby expanding our understanding of the genetics of ET. The study's findings were very meaningful, but some aspects remained unclear in the article. Here are my suggestions:

- 2.1 The study is not a multi-omics study but a GWAS, so the title of the article is misleading.

Response (2.1)

We have changed the title to **“GWAS meta-analysis uncovers novel risk loci and functional insights in essential tremor pathogenesis.”** (page 1)

- 2.2 The section on candidate gene identification mentions sQTLs but no corresponding results are presented.

Response (2.2)

We searched for sQTLs in whole blood and none were significant. This has been addressed in the revised manuscript (page 7).

“There were no sQTLs at the loci in whole blood.”

- 2.3 The familial analysis section is too simplistic. Please expand on the methods and results sections, such as the number of analyzed pedigrees and the types of variants analyzed.

Response (2.3)

The reviewer is correct. We have stated the number of families analyzed, defined what we mean by “high-impact” variants, and described the approach we used.

“...We did not find high-impact variants, including start-lost, stop-gain, stop-lost, splice donor, splice acceptor, or frameshift, segregating among 53 large families with high incidence of ET ($N \geq 5$).” (page 8)

“We used family-based method to test rare coding variants for segregation within a pedigree. We focused our search on rare (carried by <30 whole-genome-sequenced individuals), high-penetrance coding variants that could account for the familial clustering. To test for association, we created a scoring function based on the coding effect of the variant and its cosegregation with ET, inside and outside of the pedigrees, and used genome-wide simulations to estimate the significance. This method has been described in detail elsewhere². “ (page 27)

- 2.4 The genetic correlation section is similar to previously reported results. Please explain the unique aspects of this study.

Response (2.4)

The previous study tested genetic correlations with 219 phenotypes. We both extend the previous study by testing correlation with 1142 phenotypes and include a larger GWAS meta-analysis. Unlike the previous study, which tested genetic correlation with neuroticism (a personality trait), we test the correlation with anxiety-related phenotypes. This aspect is of particular relevance in the context of ET, as tremor severity in ET patients is known to increase under anxiety/stress-inducing conditions. Thus, our analysis is a broader and better powered approach.

- 2.5 The post-GWAS section highlights multiple signals indicating an association with ET for the *GCKR* gene. Please discuss the potential mechanisms of this gene in the discussion section.

Response (2.5)

We thank the reviewer for the suggestion and have added a paragraph on *GCKR* to the Discussion (page 12).

“*GCKR* has been associated with high serum uric acid and purine metabolism disorders³. Extrapyramidal signs such as tremor are often observed in these disorders⁴. Individuals with high serum uric acid might experience tremors attributed to these metabolic issues, yet, due to the lack of routine serum uric acid testing, they could be incorrectly diagnosed with ET. Whether the tremor associated with *GCKR* variant is a phenocopy or typical ET needs to be investigated further.”

- 2.6 The paper contains several errors, please carefully review and make corrections. For example, main text line 99, "frequency (MAF) over 0.01%"; figure 1, the sample size and the number of variants do not match with other parts of the manuscript; supplementary Figure 3, "for the 11 variants associated with ET."

Response (2.6)

We thank the reviewer for noticing these errors. We have read the manuscript and corrected the errors we found:

The figure legend of Supp. Fig. 3 now mentions 12 variants, and not 11.

There are a few sentences that may be misleading and we have clarified those:

We use variants with minor allele frequency over 0.01% in all datasets except the Estonian dataset and the previously reported GWAS summary statistics. We have added to Figure legend 1 (page 15) **“We included variants with MAF>0.01% in all datasets except for the Estonian dataset and the previous GWAS⁵, where variants with MAF>1% were included.”**

We also added to the main text (page 6) **“...minor allele frequency (MAF) over 0.01% in each dataset (except the Estonian dataset and the reported GWAS, where variants with MAF over 1% were included).”**

The number of variants in Figure 1 do not match Supp. Data 1 because in Figure 1 we report variants with MAF>0.01% and in Supp. Data 1 we report variants with

MAF>1%. We understand that this may be confusing to readers and have changed the Supp. Data 1 to match Figure 1.

References

1. Eggertsson, H. P. *et al.* GraphTyper2 enables population-scale genotyping of structural variation using pangenome graphs. *Nat. Commun.* **10**, 5402 (2019).
2. Steinberg, S. *et al.* Truncating mutations in RBM12 are associated with psychosis. *Nat. Genet.* **49**, 1251–1254 (2017).
3. Sandoval-Plata, G., Morgan, K. & Abhishek, A. Variants in urate transporters, ADH1B, GCKR and MEPE genes associate with transition from asymptomatic hyperuricaemia to gout: results of the first gout versus asymptomatic hyperuricaemia GWAS in Caucasians using data from the UK Biobank. *Ann. Rheum. Dis.* **80**, 1220–1226 (2021).
4. Kamatani, N., Jinnah, H. A., Hennekam, R. C. M. & Van Kuilenburg, A. B. P. Purine and Pyrimidine Metabolism. *Emery Rimoin's Princ. Pract. Med. Genet. Genomics Metab. Disord.* 183–234 (2021). doi:10.1016/B978-0-12-812535-9.00006-6
5. Liao, C. *et al.* Association of Essential Tremor With Novel Risk Loci: A Genome-Wide Association Study and Meta-analysis. *JAMA Neurol.* **79**, 185–193 (2022).

REVIEWERS' COMMENTS:

Reviewer #1 (Remarks to the Author):

No further comments, very nice paper!

Reviewer #2 (Remarks to the Author):

There are no further comments. The corrections have been completed.